# Gold Extraction from a Refractory Sulfide Concentrate by Simultaneous Pressure Leaching/Oxidation

**Juan Carlos Soto-Uribe** [1,*], **Jesus Leobardo Valenzuela-Garcia** [1,*], **Maria Mercedes Salazar-Campoy** [1], **Jose Refugio Parga-Torres** [2], **Guillermo Tiburcio-Munive** [1], **Martin Antonio Encinas-Romero** [1] and **Victor Manuel Vazquez-Vazquez** [1]

[1] Department of Chemical Engineering and Metallurgy, Universidad de Sonora, Blvd. Luis Encinas y Rosales S/N, Col. Centro, Hermosillo 83000, Mexico

[2] Institute Technology of Saltillo, Department of Metallurgy and Materials Science, Saltillo 25280, Mexico

* Correspondence: carlos.soto@unison.mx (J.C.S.-U.); jesusleobardo.valenzuela@unison.mx (J.L.V.-G.)

**Abstract:** Most gold deposits occur associated with sulphides like pyrite and arsenopyrite; thus, precious metal dissolution is possible by oxidizing auriferous sulfide concentrate using simultaneous pressure oxidation and cyanidation. The effectiveness of this process of extraction can be influenced by the temperature, cyanide (NaCN) concentration, and oxygen pressure. In this study, we conducted experiments to characterize the effects on gold extraction of ores using a range of sodium cyanide concentrations (1–8 g/L), temperatures (40–75 °C), and oxygen pressures (0.5–1.1 MPa). Characterization of the ores showed that pyrite and quartz were the main minerals present in the concentrate. The best results in terms of the highest extraction of Au were obtained with an oxygen pressure of 0.5 MPa, 6 g/L sodium cyanide, and a temperature of 75 °C, along with a constant stirring speed of 600 rpm. These conditions allowed for approximately 95% gold extraction in 90 min.

**Keywords:** cyanidation; gold-bearing; pressure leaching; pressure oxidation

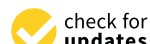



## 1. Introduction

Many gold deposits contain finely disseminated gold particles in iron sulfide minerals such as pyrite. These minerals are called refractory gold ores because of the presence of gold, which occurs either as disseminated fine gold particles—typically <1 μm [1] or as fine particles of gold encapsulated in the ore host [2,3]. In certain ores, extraction rates corresponding to lower than 60% gold recovery occur when using direct cyanide leaching, even after grinding, hence indicating that the concentrate is refractory. Thus, a suitable pretreatment process is often required to make cyanidation more effective. The refractory ore must be destroyed by chemical means using oxidative processes such as oxidation by roasting [4,5], bio-oxidation [6–9], pressure oxidation [10,11], or ultrafine grinding [12,13].

The Velardeña mine is in Durango State, wherein the gold concentrate is produced by sulfide flotation and sold to external processing companies. Processing of this gold concentrate is generally conducted using roasting and leaching methods. However, pressure oxidation has been used to treat refractory gold ores or concentrates as pretreatment and subsequent cyanidation. Alkaline pressure oxidation is an effective pretreatment for releasing refractory gold from pyrite and arsenical concentrates before cyanidation [11,14]. Simultaneous pressure leaching oxidation involves the oxidation of pyrite and Au cyanidation in alkaline media, where oxidation liberates or exposes gold particles that are otherwise occluded or finely disseminated. In addition, the dissolution of gold is also dependent on oxidation due to its electrochemical behavior, which can be represented by the Elsner equation [1]. The equation proposed for gold is:

$$4Au + 8CN^- + O_2 + H_2O \rightarrow 4\left[Au(CN)_2^-\right] + 2OH^- \qquad (1)$$

The pressure oxidation of pyrite involves reactions yielding ferrous ions, sulfate ions, and elemental sulfur as products [11,15–18]. Pyrite oxidation in alkaline media is described by Equation (2), where the reaction is similar to acid oxidation (Equations (2)–(6)) [1]. The reactions are shown as follows:

$$2FeS_2 + 7.5O_2 + 7H_2O \leftrightarrow 2Fe(OH)_3 + 8H^+ + 4SO_4^{2-} \qquad (2)$$

$$2FeS_2 + 7O_2 + 2H_2O \rightarrow 2FeSO_4 + 2H_2SO_4 \qquad (3)$$

$$2FeSO_4 + H_2SO_4 + 0.5O_2 \rightarrow Fe_2(SO_4)_3 + H_2O \qquad (4)$$

$$Fe_2(SO_4)_3 + 3H_2O \rightarrow Fe_2O_3(\downarrow) + 3H_2SO_4 \qquad (5)$$

$$4FeS_2 + 15O_2 + 8H_2O \rightarrow 2Fe_2O_3 + 8H_2SO_4 \qquad (6)$$

Several authors have published their results from gold extraction via oxidative pretreatment following the cyanidation process [14,19–21]. In our study, we evaluate the effect of the simultaneous application of pressure leaching and cyanidation on the rate of gold dissolution from gold sulfide concentrate.

## 2. Materials and Methods

A pyrite gold concentrate was provided by La Velardeña Mine. The sample was homogenized and screened using 100 Tyler mesh (100% passing size). X-ray diffraction (XRD), scanning electron microscopy (SEM), and energy-dispersive X-ray spectroscopy (EDX) were used to obtain characteristics of the concentrate for application in combination with physical characterization. Chemical analysis (CA) was performed using the fire assay technique to determine the gold grade, and atomic absorption spectroscopy (AAS) was conducted to determine the presence of copper, zinc, and iron.

The sample was then subjected to metallurgical tests of simultaneous pressure leaching/oxidation in media alkaline. The tests were conducted in a 1 L Parr pressure reactor with a heating jacket, stirrer, and inlet and outlet gas control valves (Figure 1) under the following conditions: a solid ratio of 20%, pressure range of 0.5–1.1 MPa, temperature range of 45–75 °C, sodium cyanide concentration of 4–8 g/L, pH of 11–12 using CaO, and a reaction time of 90 min. The experimental variables and parameters for pressure leaching/oxidation are given in Table 1.

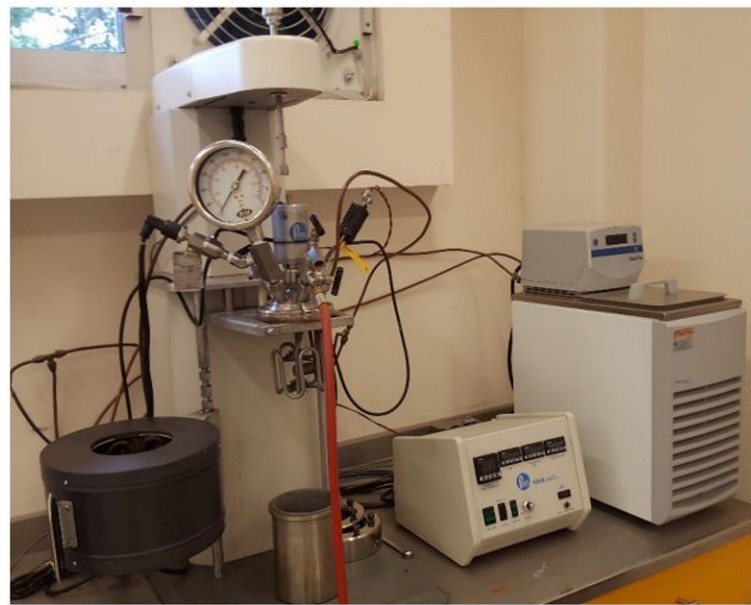

**Figure 1.** Experimental setup for simultaneous pressure leaching and oxidation.

**Table 1.** Process variables and parameters for simultaneous pressure leaching and oxidation.

| Parameter | Value |
|---|---|
| Oxygen pressure, MPa | 0.5, 0.8, and 1.1 |
| Temperature, °C | 45–75 |
| NaCN, g/L | 4–8 |
| Solids, wt.% | 20 |
| Stirring speed, rpm | 600 |

For conventional cyanidation, the tests were conducted at room temperature and under atmospheric pressure conditions: 2 g/L NaCN, 3 kg/ton CaO for pH control, a solid ratio of 20%, 72 h leaching time, and particle size of −100 mesh (149 μm).

## 3. Results and Discussion

### 3.1. Mineralogy Characterization

The scanning electron microscopy (SEM) results in Figure 2a) indicate the existence of acanthite resulting from the high percentage of Ag and S, and it is also possible to observe encrusted quartz impurities and galena in a pyrite matrix. In the X-ray diffraction (XRD) pattern in Figure 2b), the spectrum exhibits higher crystallinity for pyrite ($FeS_2$), 81.3%, and quartz ($SiO_2$), 11.6%, with both being the main species, thus confirming the concentrate's refractory nature. Other species such as calcite ($CaCO_3$) and galena (PbS) are present in minor quantities.

The Au and Ag grades in the head concentrate and other metals were determined by CA (Table 2). Table 3 shows the mineralogical reconstruction via XRD, and CA is expressed in terms of the weight percentage (wt.%).

**Table 2.** Chemical head grade analysis through fire assay (FA) and 4-acid digestion with atomic adsorption finish.

| Mineral Concentrate Sample | | | | | | | | |
|---|---|---|---|---|---|---|---|---|
| Au, g/t | Ag, g/t | Cu, % | Zn, % | Fe, % | Pb, % | As, % | S, % | Insol., % |
| 42.2 | 3932 | 2.1 | 4.1 | 32.7 | 3.3 | 0.17 | 31.2 | 23.3 |

**Table 3.** Mineralogical reconstruction of the gold concentrate.

| Compounds | | Weight % |
|---|---|---|
| Chalcopyrite | $CuFeS_2$ | 4.4 |
| Galena | PbS | 3.03 |
| Sphalerite | ZnS | 4.8 |
| Pyrite | $FeS_2$ | 64.2 |
| Arsenopyrite | FeAsS | 0.38 |
| Quartz | $SiO_2$ | 19.8 |
| Calcite | $CaCO_3$ | 3.3 |

### 3.2. Conventional Cyanidation Test

Low extraction rates were observed for the cyanidation test at 30 °C and atmospheric pressure. A conventional cyanidation test was carried out under the following conditions: leaching time of 72 h and particle size of −100 mesh (149 μm), 2 g/L NaCN concentration, the addition of 3 kg/t CaO to control pH, and 20% solids.

As shown in Figure 3, conventional cyanide leaching results in a low extraction of 46%, which may be associated with the use of a sulfide refractory gold ore. Salazar-Campoy et al. [22] obtained 37.9% Au extraction in 72 h from sulfide refractory ore, whereas Elorza-Rodriguez et al. [23]

recovered 61.2% Au following standard cyanidation for refractory ores. Figure 3 shows that the Au extraction using conventional cyanidation was lower than 50%, in contrast to reports involving simultaneous pressure leaching and oxidation in [18,22], where >90% gold was recovered.

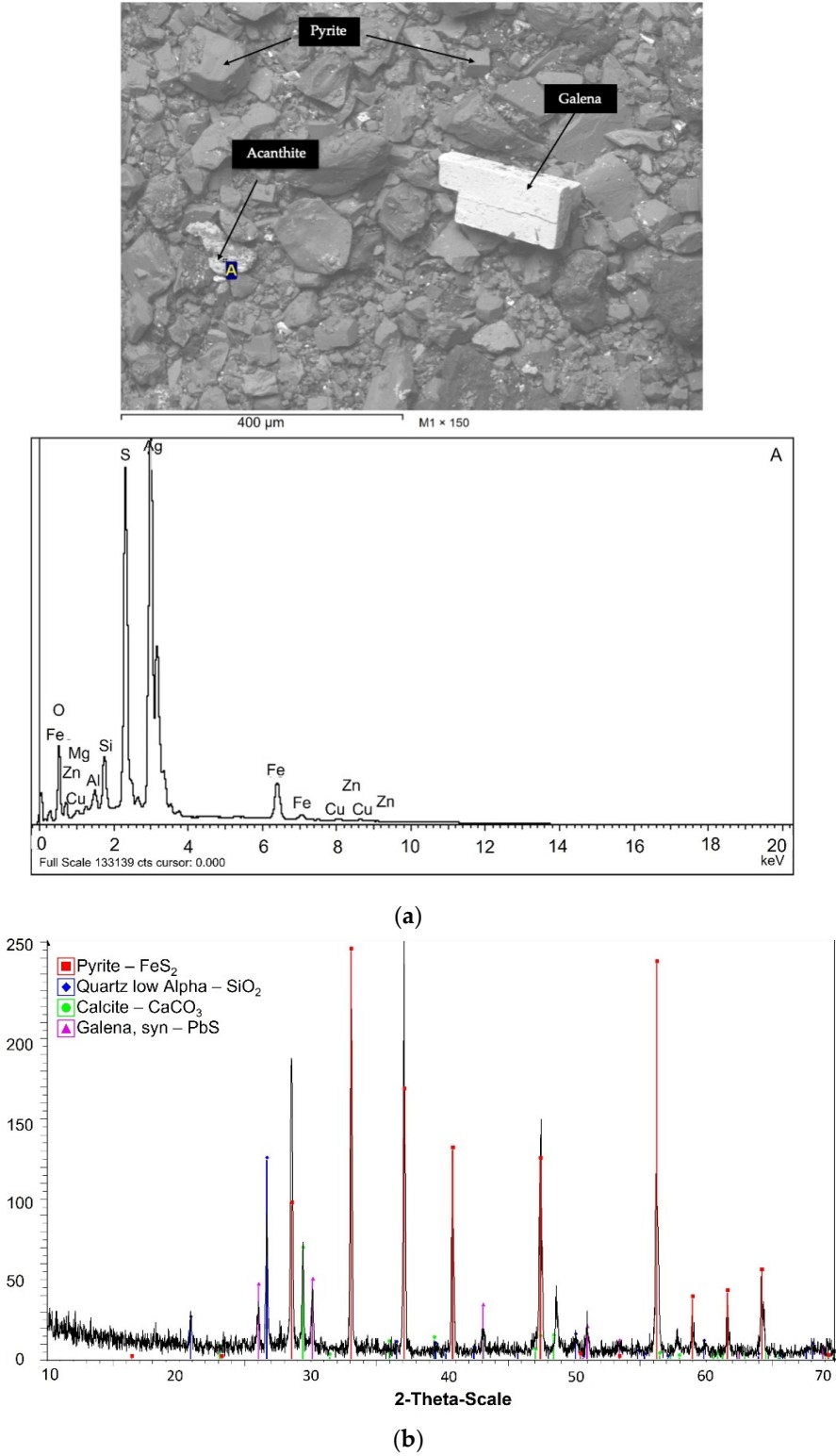

**Figure 2.** (**a**) Scanning electron microscopy of the sample at 400 μm showing pyrite, argentite, and galena particles, and energy dispersive X-ray spectroscopy images of particles. (**b**) X-ray diffraction pattern of concentrate sample.

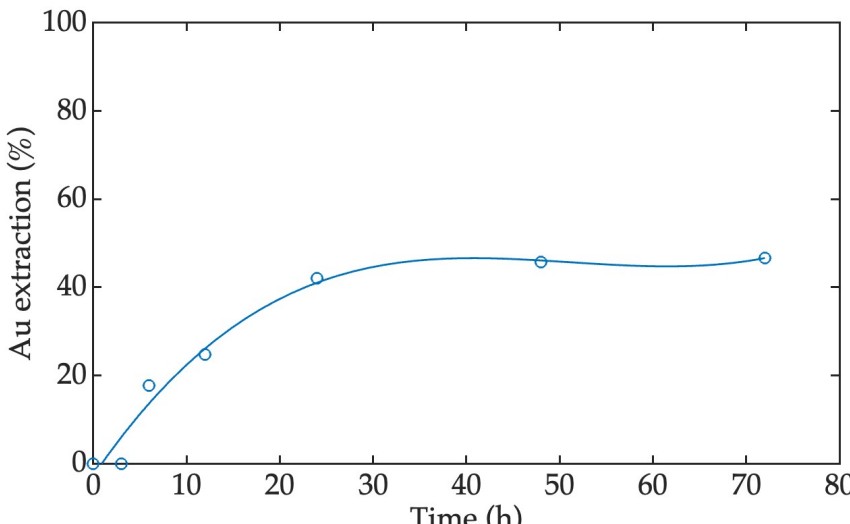

**Figure 3.** Au recovery (%) resulting from cyanidation at room temperature and atmospheric pressure as a function of time.

### 3.3. Simultaneous Pressure Leaching and Oxidation

Oxidation and pressure leaching tests were performed to determine temperature, oxygen pressure, and cyanide concentration effects on gold extraction. The tested variations were temperatures ranging from 45–75 °C, oxygen pressure from 0.5 to 1.1 MPa, and sodium cyanide concentrations in the range of 4 to 8 g/L.

### 3.3.1. Effects of Temperature and Pressure on Gold Extraction

Results for gold extraction from oxidation in the leaching test at a temperature of 40–75 °C and with simultaneous pressure increases are shown in Figure 4. For the range of tested pressures and 6 g/L of NaCN, the Au (%) extracted increased due to increasing oxygen pressure, whereas the temperature raise, recovery of Au (%) decreased at low pressures due to the effects on dissolved oxygen, which decreases according to Henry's law. Rusanen et al. [11] argue that oxygen pressure is the most important factor influencing the oxidation of refractory minerals.

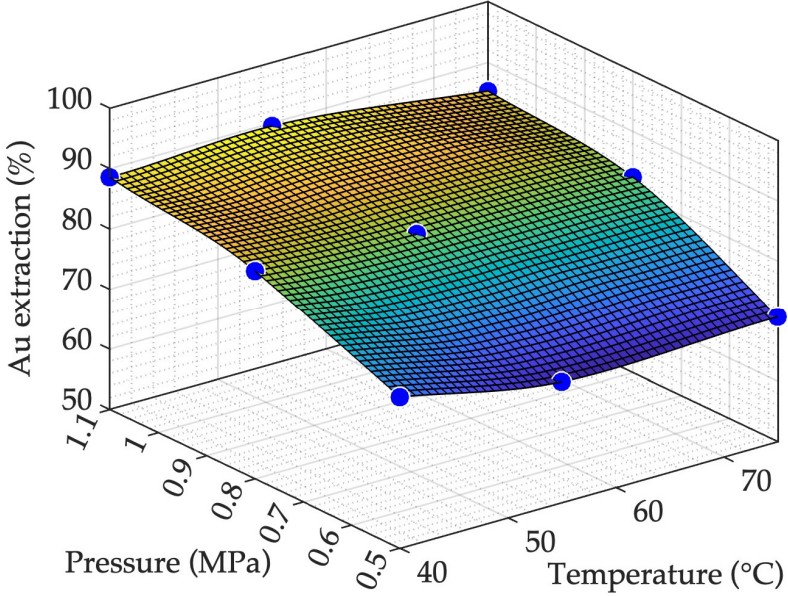

**Figure 4.** Effects of temperature and pressure on Au extraction at 6 g/L cyanide concentration.

From Figure 4, we can see that gold dissolution of 89.51% was obtained at pressure leaching and oxidation conditions of 1.1 MPa and 55 °C. Leaching and oxidation under increased pressure enhance gold extraction compared with tests conducted under atmospheric conditions.

The effects of temperature on Au extraction were studied, as shown in Figure 5. Generally, the increase in temperature also results in increased amounts of extracted gold. Nevertheless, compared with conventional cyanidation, simultaneous pressure leaching oxidation results in higher Au extraction within a short time, obtaining 41%, 54%, and 82% for each temperature in 10 min. Azizi et al. [24] indicated that the presence of galena during the cyanidation of pyrite ore increases the recovery of gold from 2% to 58%.

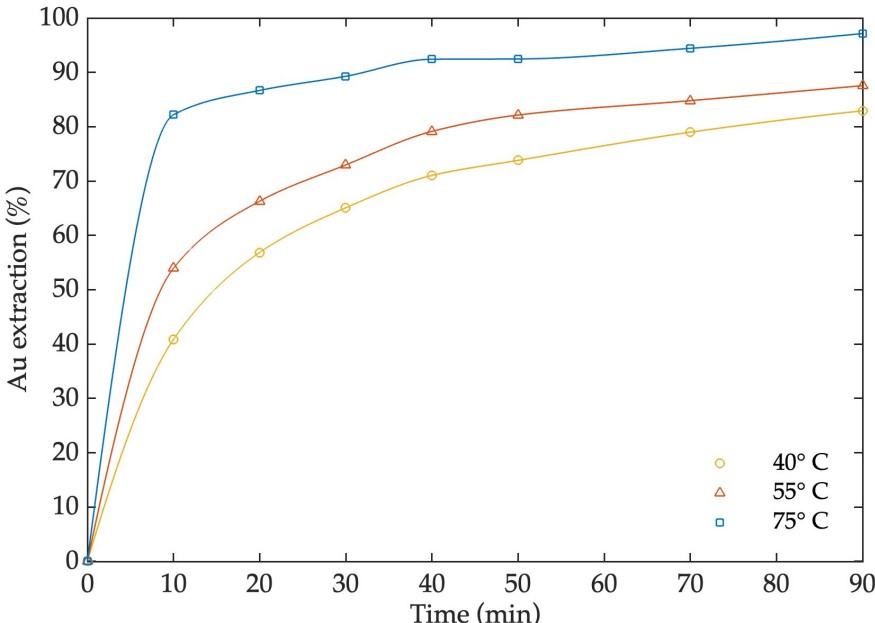

**Figure 5.** Au extraction (%) from simultaneous pressure leaching and oxidation at a pressure of 0.5 MPa and 6 g/L cyanide concentration.

According to Thomas [16], an increase in temperature will, in nearly all cases, increase the rate of a chemical reaction to a significant extent: for every 10 °C rises in temperature, the specific rate is increased by a factor of two or three. In a process similar to the chlorination leaching of pressure oxidized for refractory gold concentrate, the maximum temperature of 50 °C decreases the extraction due to the decomposition of agent leaching [25].

### 3.3.2. Effect of Cyanide Concentration

Sodium cyanide concentration (4–8 g/L) has an insignificant effect on gold extraction, as shown in Figure 6. According to Fleming [2], the cyanide concentration is less important than the oxygen concentration in gold extraction.

Results show that the use of excessive cyanide (above the optimum level) usually results in unnecessary cyanide consumption and has no beneficial effect on gold extraction. However, a high concentration of free cyanide is required for effective gold leaching because of the nature of the ore, which contains soluble sulfides. Cyanide consumption occurs because of the presence of other metal ions, such as copper and other metal ions. In this sulfide concentrate, the amount of copper is low (2.1%), whereas copper is very slightly dissolved [26]. In addition, at the temperature range of 45–75 °C, cyanide is not degraded. Parga et al. [18] have reported that gold recovery decreases at temperatures above 80 °C due to the oxidation of cyanide ions.

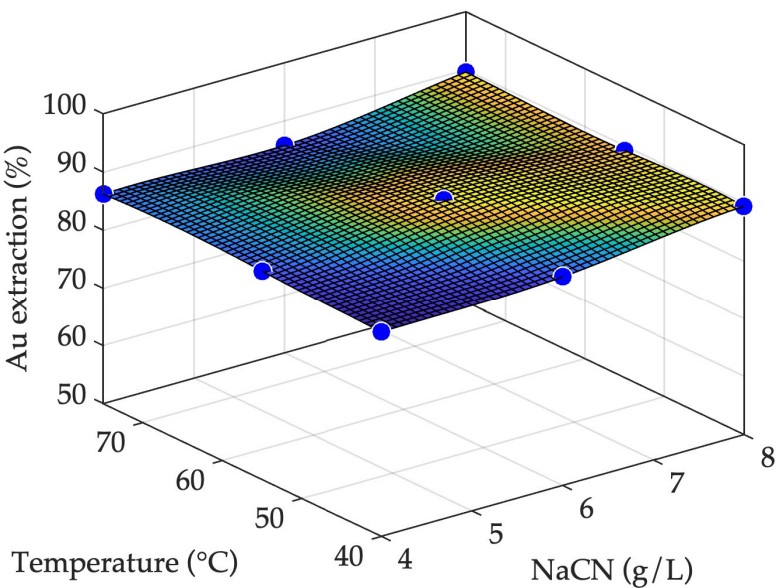

**Figure 6.** Effects of cyanide concentration on Au extraction (%) during simultaneous pressure leaching and oxidation at a pressure of 0.8 MPa.

### 3.3.3. Effect of Particle Size on Gold Extraction

The gold concentrate was ground and screened at particle sizes of +173, −173/+147, −147/+104, −104/+74 μm to carry out tests using 6 g/L NaCN concentration, 0.5 MPa oxygen pressure, and 40 °C for 90 min of simultaneous pressure leaching and oxidation. Figure 7 shows that the particle size has an important effect on gold extraction, wherein the extraction of Au increases for small particle sizes. This effect was described by Chaidez et al. [27] for Cu extraction from chalcopyrite concentrate using acid leaching at low pressure.

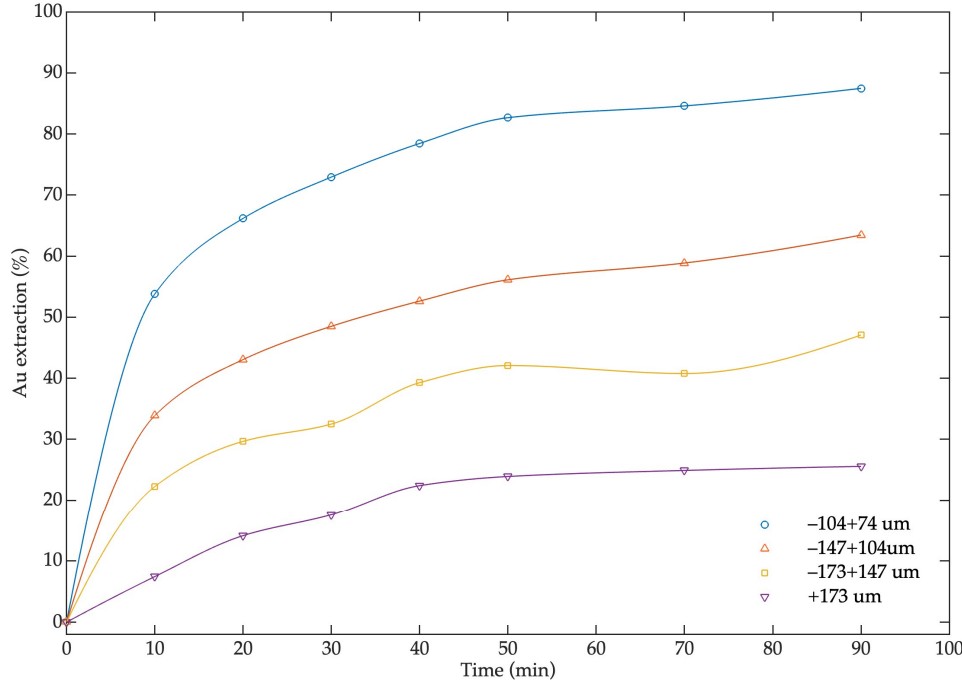

**Figure 7.** Effect of particle size (micron, μm) on Au extraction at 6 g/L cyanide concentration, 0.5 MPa pressure, and 40 °C.

### 3.3.4. Simultaneous Pressure Leaching and Oxidation

During simultaneous pressure leaching and oxidation, the gold concentrate is oxidized and leached in a single stage. Pyrite was present in a high amount in the concentrate in this study; it was oxidized in alkaline media due to NaCN working at a pH range of 11 to 12, Shown in Figure 8a) are XRD results for a sample of gold concentrate produced by alkaline oxidation via simultaneous leaching and oxidation at 75 °C and 1.0 MPa oxygen pressure. X-ray mapping of the pyrite particles is shown in Figure 8b).

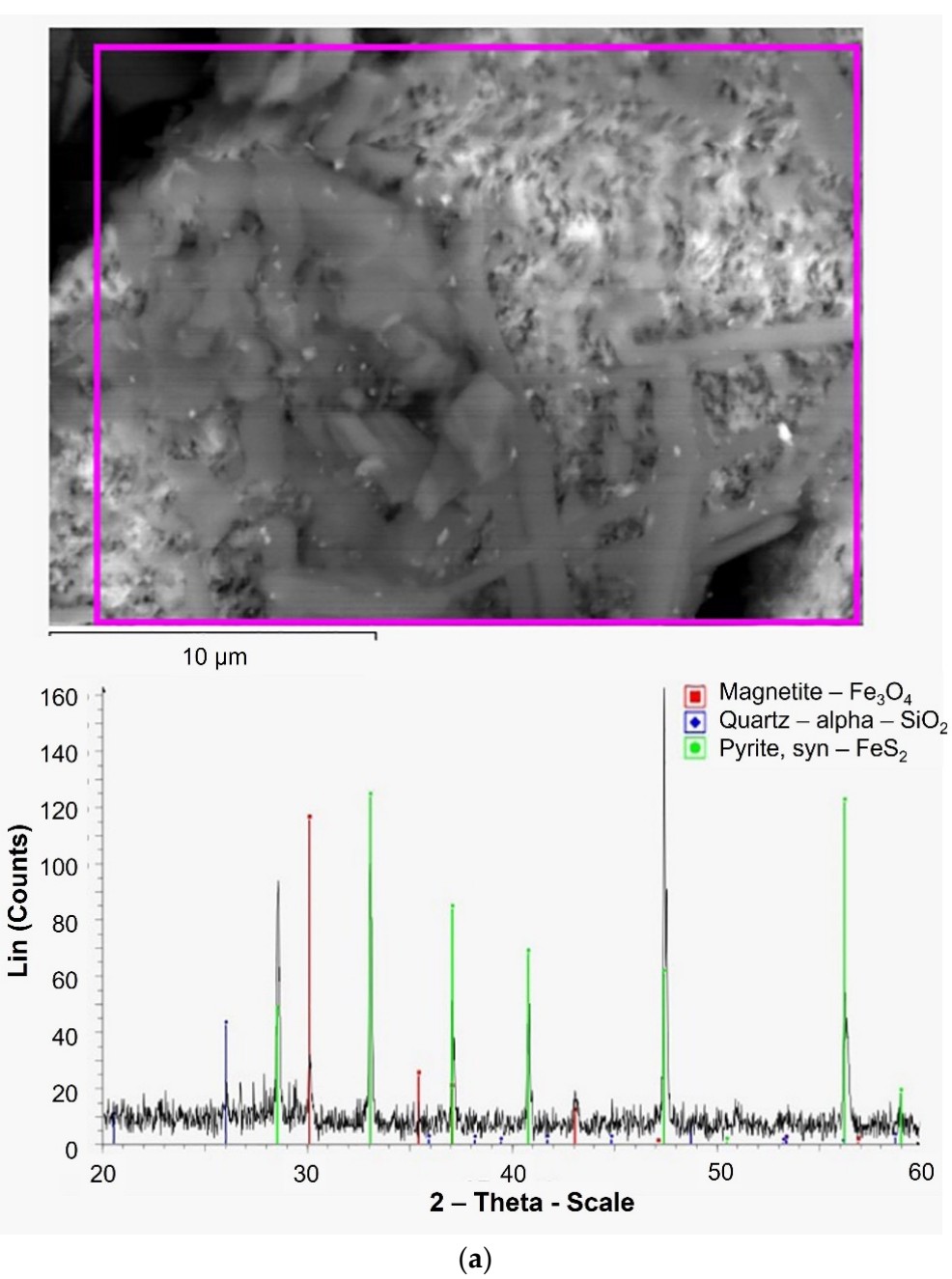

(a)

**Figure 8.** *Cont.*

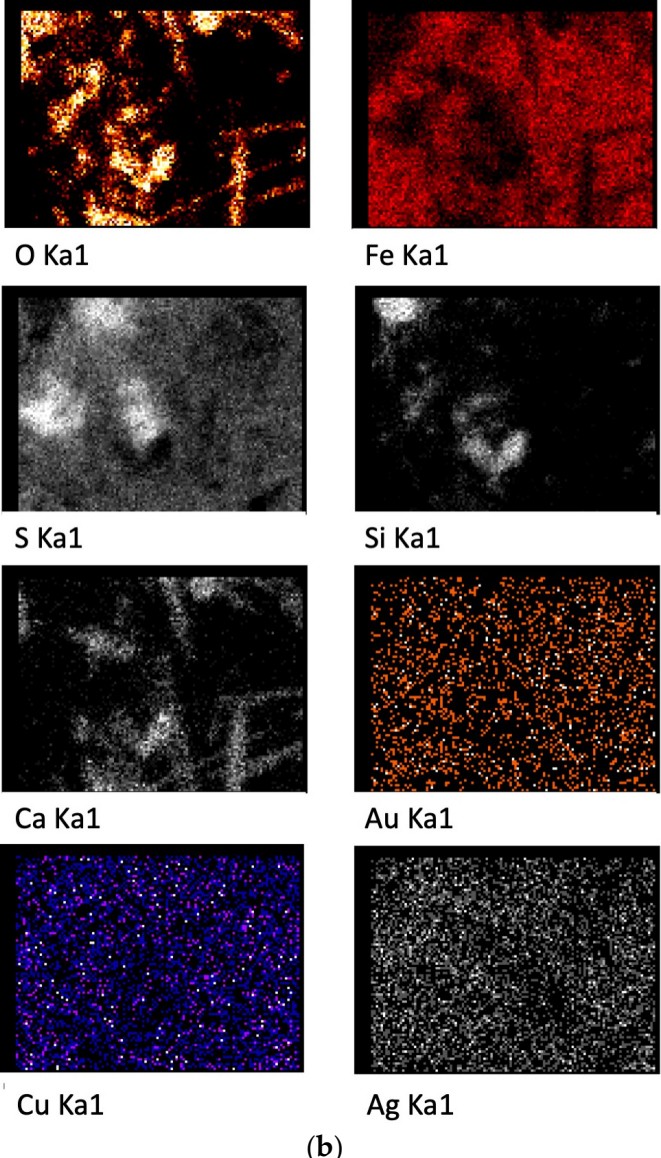

O Ka1          Fe Ka1

S Ka1          Si Ka1

Ca Ka1          Au Ka1

Cu Ka1          Ag Ka1

(**b**)

**Figure 8.** (**a**) Punctual microanalysis of particles by SEM–EDS micrograph showing a pyrite particle at 10 μm, and X-ray diffraction pattern of oxidized gold concentrate. The spectrum indicates the presence of pyrite, quartz, and magnetite. (**b**) X-ray mapping of the pyrite particles tests by SEM-EDS.

Pyrite alkaline oxidation results in hematite production. Caldeira et al. [15] indicated that the oxidative leaching of pyrite produces an insoluble iron oxide precipitate, and hematite was found in the precipitate, similar to in the main phase. Hematite is porous, and this improves the gold recovery via leaching. The magnetite in Figure 8 indicates partial oxidation of pyrite by low oxygen concentration, due the low oxygen pressure used on test.

## 4. Conclusions

The low amount of gold extracted by conventional cyanidation (46%) occurs when the mineral has refractoriness properties due to occlusion by the pyrite matrix. When the pressure leaching and oxidation during processing were 1.1 MPa and 55 °C, a gold dissolution of 89.51% was achieved. Pressure leaching and oxidation enhance gold extraction compared with tests conducted under atmospheric conditions. The effect of particle size on gold dissolution indicates that Au(%) extraction was increased with decreased particle size, 87–88% Au extraction was achieved at a size fraction of −104/+74 μm.

**Author Contributions:** Conceptualization, J.C.S.-U. and J.L.V.-G.; formal analysis, J.C.S.-U. and M.M.S.-C.; investigation, J.C.S.-U., J.L.V-G and G.T.-M.; methodology, J.C.S.-U., J.L.V.-G and J.R.P.-T.; project administration, J.L.V.-G. and G.T.-M.; resources, V.M.V.-V., M.A.E.-R. and J.L.V.-G.; data curation, G.T.-M., M.A.E.-R. and M.M.S.-C.; formal analysis, J.C.S.-U. and J.L.V.-G.; supervision, J.L.V.-G. and M.M.S.-C.; validation, J.C.S.-U., M.A.E.-R. and V.M.V.-V.; writing—original draft, J.C.S.-U.; writing—review and editing, J.L.V.-G., J.R.P.-T., M.M.S.-C. and V.M.V.-V. All authors have read and agreed to the published version of the manuscript.

**Funding:** This Research received no external funding.

**Acknowledgments:** The authors gratefully acknowledge CONACYT for graduate scholarship to one author (J.C.S.U.), National Laboratory of Geochemistry and Mineralogy, Engineering Division and the Department of Chemical Engineering and Metallurgy of the University of Sonora and TNM—Institute of Technology of Saltillo.

**Conflicts of Interest:** The authors declare no conflict of interest.

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
