# Peer review of "Gold Extraction from a Refractory Sulfide Concentrate by Simultaneous Pressure Leaching/Oxidation"

_minerals, doi:10.3390/min13010116_

Round 1
Reviewer 1 Report
Abstract
Line 11 …outside of sulphides..(?) should be ‘….associated with sulphides’?
Line 20 refers to 99% silver extraction, however extraction of silver is not discussed in the paper.
Line 29 .. the ore is refractory.
Figure 2 – the diagram is too small to be legible. In any case the figure is not needed because it shows analytical methods that are widely used and understood. It is presumed that XRD and SEM/EDX have been interpreted correctly.
Line 48 – alkaline media?
Line 84: Section 3.1 Mineralogy characterization. Refractory gold ores can contain particulate gold as inclusions in pyrite, or as solid solution in pyrite. The latter can only be detected by using techniques such as SIMS or LA-ICP-MS. The difference in the two forms can be significant in interpreting test results in some ores (e.g., solid solution does not respond to finer milling). It would make the paper more complete if pyrite was analysed for solid solution gold.
Line 86 – the correct mineral name for Ag2S is acanthite, not argentite. Argentite is the high T form, but all low T Ag2S is acanthite.
Figure 4: data points appear to be at 0.5, 0.8 and 1.1 MPa; however, Table 1 states that oxygen pressure is measured at 0.5, 1.0 and 1.1 MPa.
Line 179 – should be -174/+147
Figure 7: should be +174 in figure caption
Line 191: section 3.3.4. Refers to paper by Caldeira et al (2003) that states that haematite is the oxide produced by oxidation of pyrite in alkaline cyanide solution. However, XRD in Figure 8 indicates pyrite, quartz and magnetite, which is inconsistent with results of Caldeira et al (2003). This discrepancy should be explained.
Figure 8: what is punctual microanalysis? Figure too small to see detail. Interpretation of x-ray maps is questionable. Quartz is evident (Si and O). High S appears to correspond to low Fe – not pyrite? Ca and O appear to indicate calcite (not mentioned in caption). Cu, Au and Ag maps show nothing.
Line 214: …was 83 wt% at -74 μm ?? not sure what is meant by this.
Author Response
Reviewer 1 Response Letter
Article: Gold extraction from a refractory sulfide concentrate by simultaneous
pressure leaching/oxidation.
The electronic copy of the revised manuscript is being returned to you with changes suggested by the reviewers. The very insightful reviewers’ comments are well taken and are responded to accordingly.
We have reviewed all the comments and the corrections are incorporated into the paper. By making the above additions/revisions to the manuscripts we hope to have properly responded to the reviewers’ comments and suggestions. Thank you very much for considering our manuscript and your assistance in having it published. We would like to continue working with you for our future research publications.
Best regards.
Reviewer 1 Response Letter
Article: Gold extraction from a refractory sulfide concentrate by simultaneous
pressure leaching/oxidation.
The electronic copy of the revised manuscript is being returned to you with changes suggested by the reviewers. The very insightful reviewers’ comments are well taken and are responded to accordingly.
We have reviewed all the comments and the corrections are incorporated into the paper. By making the above additions/revisions to the manuscripts we hope to have properly responded to the reviewers’ comments and suggestions. Thank you very much for considering our manuscript and your assistance in having it published. We would like to continue working with you for our future research publications.
Best regards.

Reviewer 2 Report
Authors investigated gold extraction from a refractory sulfide concentrate by using simultaneous pressure oxidation and cyanidation. A gold dissolution of 89.51% was achieved when the pressure leaching and oxidation during processing were 1.1 MPa and 55 °C. However, it was only 46% by conventional cyanidation. The following corrections/revisions essentially required to be addressed as given below:
(1) In Line 20, the extraction of approximately 95% gold and 99% silver cannot be found in the Results and Discussion.
(2) The measuring error of gold concentration?
(3) The size distribution of the refractory sulfide concentrate?
(4) For conventional cyanidation, CaO was used for pH control. What is the alkaline medium for simultaneous pressure leaching/oxidation?
(5)The sulfur concentration is not given.
(6)In Page 5, the increases in temperature decreased recovery of Au (%) at low pressures for the range of tested pressures and 6 g/L of NaCN. In Page 6, the increase in temperature also result in increased amounts of extracted gold. Temperature has a positive or negative correlation with gold extraction?
(7)In Line 179, It should be -174/+147.
(8)In Section 3.3.3, the effect of particle size of -74 μm is not given. In practice, the particle size of -74 μm accounts for larger than 90%.
(9)The effects of solid ratio and stirring speed are not given.
(10) The optimized parameters and corresponding gold extraction are not given.
Author Response
Reviewer Response Letter
Article: Gold extraction from a refractory sulfide concentrate by simultaneous
pressure leaching/oxidation.
The electronic copy of the revised manuscript is being returned to you with changes suggested by the reviewers. The very insightful reviewers’ comments are well taken and are responded to accordingly.
We have reviewed all the comments and the corrections are incorporated into the paper. By making the above additions/revisions to the manuscripts we hope to have properly responded to the reviewers’ comments and suggestions. Thank you very much for considering our manuscript and your assistance in having it published. We would like to continue working with you for our future research publications.
Best regards.

Round 2
Reviewer 2 Report
The following corrections/revisions essentially required to be addressed as given below:
(1) In Line 18-21, the best results in terms of the highest extraction of Au were obtained with an oxygen pressure of 1.0 MPa, sodium cyanide, and temperature of 55 °C, along with a constant stirring speed of 600 rpm. These conditions allowed for extraction of approximately 95% gold in 90 min. However, the extraction of approximately 95% gold in 90 min could only be found in Figure 5 with an oxygen pressure of 0.5 MPa, 0.6 wt % cyanide concentration, and temperature of 75 °C.
(2) The sulfur concentration is still not given in Section 2 Materials and Methods or Section 3.1. Mineralogy characterization.
(3) The alkaline medium for simultaneous pressure leaching/oxidation should be added in Section 2 Materials and Methods
(4) However the reactions of Chalcopyrite, Galena, Sphalerite, Arsenopyrite during the simultaneous pressure leaching and oxidation?
(5) During the simultaneous pressure leaching and oxidation, CaSO4 could be obtained due to the reaction between CaO and newly generated sulfate ion, which affects the extraction of gold. How reduces this side effect?
